# Integrated Environmental Effect Assessment on Large Coal-Electricity Production Base and Sustainability Strategy: A Case in Xilin Gol, China

**Chenxing Wang [1],\*, Yan Yan [1,2], Jiaxun Li [3], Yuan Quan [1], Shanlin Li [4] and Gang Wu [1,2]**

1   State Key Laboratory of Urban and Regional Ecology, Research Center for Eco-Environmental Sciences, Chinese Academy of Sciences, Beijing 100085, China; yyan@rcees.ac.cn (Y.Y.); yuanquan@rcees.ac.cn (Y.Q.); wug@rcees.ac.cn (G.W.)

2   College of Resources and Environment, University of Chinese Academy of Sciences, Beijing 100049, China

3   Policy Research Center, Ministry of Housing and Urban-Rural Development, Beijing 100835, China; jxli@rcees.ac.cn

4   National Science Library, Chinese Academy of Sciences, Beijing 100190, China; lishanlin@mail.las.ac.cn

\*   Correspondence: cxwamg@rcees.ac.cn; Tel.: +86-10-62849510

**Abstract:** Coal-electricity production base (CEPB) is an integrated plant comprising of coal mining, power conversation and chemical treatment in a particular area. As a representative of the national energy structure's adjustment and development, a CEPB could support the West-to-East Gas Transmission Project and manage electric networks. However, the national large-scale CEPBs are always distributed in arid and semi-arid regions in the north of China, which has already led to ecological and environmental problems to local areas. This paper aims at making a comprehensive assessment on CEPBs by building an assessment index system and an assessment model of the environmental effects of the large CEPB in Xilin Gol, China. The results showed that the integrated assessment result is 0.64, which reflects that the exploitation and construction of CEPBs have not severely damaged the ecological environment of Xilin Gol yet, but the water environment has been seriously disturbed and the damage to the atmosphere, soil and ecological system, as well as other potential ecological harm, cannot be ignored. Towards future sustainability, this paper provides five sustainable strategies which are established on a comprehensive framework, establishing monitoring mechanisms of the environmental impact process, devising metrics of sustainability and keeping regular assessment using fine management to continuously decrease the pollutants in production processes and meet goals of regional sustainable development.

**Keywords:** coal-electricity production base (CEPB); information entropy; analytic hierarchy process (AHP); environmental effects; sustainability strategy

## 1. Introduction

Coal-electricity production base (CEPB) is an integrated base that aims at realizing full power production in a coal base and transmitting power outward. A group of power bases will be constructed in CEPBs according to the output and reserve of available coals that could be used to generate electricity in the area. A CEPB provides basic energy support for the West-to-East Gas Transmission Project and electric network, which has far-reaching influence on the national energy use pattern and the development of the coal industry [1,2]. A CEPB transports clean energy to North China, as well as Northeast and East China urban agglomerations. However, the effect on the ecological environment in CEPB areas has been increasing, which has become the key issue of CEPBs' sustainable development [3]. Based on the study of

major problems emerging in the region, taking effective measures to improve the ecological environment of a CEPB has been considered to be the key solution.

According to China's coal industry five-year plan (2014–2020), China has planned and constructed nine CEPBs in Xinjiang, Inner Mongolia, Shanxi, Ningxia, Shaanxi. etc., which are all in arid and semi-arid regions. These areas mainly belong to temperate desert and temperate grassland ecosystems, and these ecosystems are so vulnerable that they can be easily damaged by environmental problems such as vegetation degradation, minimal evaporation and soil erosion [4]. Now China has completed the construction of CEPBs, but the process of construction may have already caused more serious ecological and environmental problems, such as grassland degradation, soil desertification, soil erosion, water and air pollution, etc. [5,6]. The coal resources are distributed unevenly over China, and the majority are found in the west and north of China. The coal resource centers are located so far away from the consumption centers that the pressure of long-distance conveyance is extremely high.

The CEPBs are located mainly in the arid and semi-arid savannas and deserts which have vital ecological functions. In view of their eco-environmental characteristics, these areas show high ecological and environmental fragility, weak capacity for ecological restoration and high trends of ecosystem degradation [7,8]. For one thing, these areas show high ecological sensitivity, including soil erosion and desertification [9]. Some CEPBs, such as Shanxi, Shanbei, Ningdong and Ordos, are in regions with high ecological sensitivity, low vegetation coverage, serious water loss and soil erosion. For another, almost all CEPBs are in desertification regions with strong winds, sandy soil and fragile ecosystems [10]. Most of all, these areas are important ecological protective screens for city groups in their east, such as Beijing-Tianjin-Hebei, the Northeast Plain, and the Yellow River Plain [11]. Therefore, the assessment of regional environmental effects and other integrated effects along with understanding the main limiting factors would greatly help CEPB sustainable development.

For better understanding of the relationship among ecosystem services, processes and construction, and operation of CEPBs, several researches have been carried out to assess the impact on ecosystem health and to pursue effective measures to ensure the sustainability of CEPBs [12]. Most of them are conducted in CEPBs by reputable research groups in China. This research covers various aspects of sustainability and strategies for CEPBs, and their findings provide important information on how to protect the ecosystem and environment associated with CEPBs [13]. The research also includes spatial distribution and development strategies of CEPBs in China, sustainability of CEPBs [14], ecological carrying capacity, ecological sensitivity and ecological compensation [15,16], as well as vegetation, soil and atmospheric environments [17,18], and ecological and environmental monitoring [5,19].

In the assessment of the environmental effect of CEPBs, the traditional analytic hierarchy processes (AHP) are often used to establish the assessment index system, and expert scoring or regional situation assessments are used to establish the weight matrix of the assessment index system. However, the construction of CEPBs involves many factors which require a complex index system and large database, and the result is not sensitive to the change of indices, either. While with the method is based on information entropy and comprehensive analytic hierarchy process, the assessment index system is established on the information entropy model, which has a small data requirement and the result is sensitive to the change of indices.

In this study, the assessment framework of the environmental effect in China's large-scale CEPB areas is established on information entropy and comprehensive AHP. The comprehensive environmental effects of the large-scale CEPB in Xilin Gol, Inner Mongolia is quantitatively evaluated. Meanwhile, the effects on ecological environment, society and economy, as well as environmental management and sustainability, are also discussed. On this basis, main factors that limit China's large CEPB sustainable development are identified, and corresponding macro-measures and sustainable development strategies are proposed.

## 2. Materials and Methods

### 2.1. Study Area

This CEPB is located in the northeast of Xilin Gol, the center of Inner Mongolia. The CEPB covers the city of Xilinhot, counties of Abag Banner, East Ujimqin Banner (including the Ulagai administrative area) and West Ujimqin Banner (Figure 1). The total area is 114,364 km², where the coal mining area is about 3340 km², and about 90% of the area is grassland. The area of grassland has been decreasing since 2000, and the degradation is serious in the recent two decades [11].

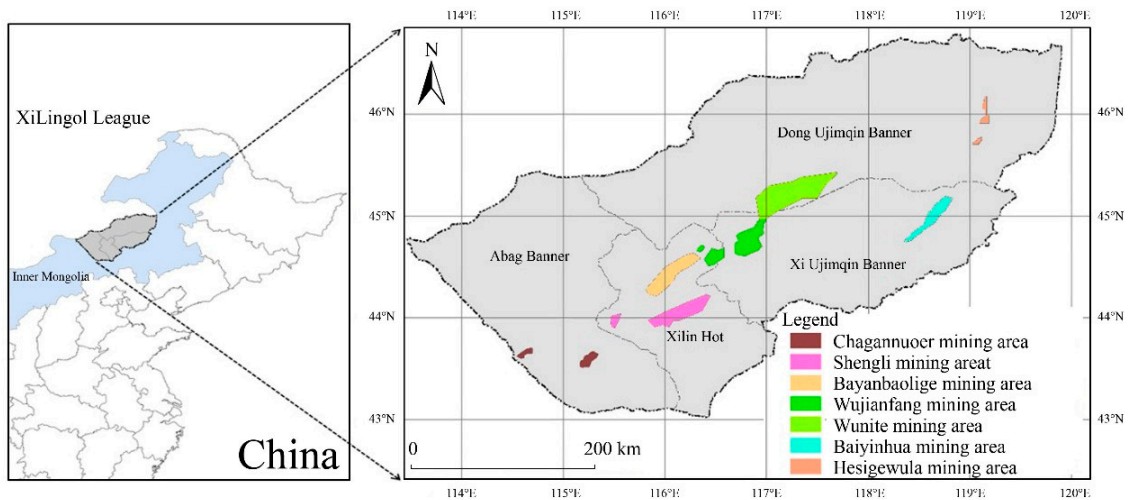

**Figure 1.** Location of study area.

The Xilin Gol CEPB contains seven comprehensive development mining areas, including Chagannuoer mining area, Wujianfang mining area, Hesigewula mining area, Shengli mining area, Wunite mining area, Baiyinhua mining area and Bayanbaolige mining area, as well as a series of pithead power clusters, coal gangue power plants and heat and power plants. The scope of the assessment includes key areas of the Xilinhot–Ujimqin coal-fired power base. The CEPB is rich in coal resources, with proven reserves containing 114.8 billion tons of coal, including five coalfields with reserves over 10 billion tons and 21 coalfields with 1–10 billion tons. The main mining method is open-pit mining. The construction of the CEPB has finished, and the mining area as well as the heat and power plants are in normal operation. The comprehensive assessment of the environmental effects in the study area could help to understand the influence of the exploitation, construction and operation on the regional ecological environment.

### 2.2. Assessment Method

The environmental effect assessment of a CEPB is mainly based on the index system and appropriate model. In traditional AHP, the system issues are simplified and hierarchy structure is established [20]. The weight of each index is calculated by solving eigenvalues and eigenvectors of the judgment matrix [21,22]. The environmental effect assessment of a CEPB involves many factors and indices, which make it difficult to acquire data and require a large index system, and the result is not sensitive to change of indices [23]. The method based on information entropy and comprehensive AHP establishes the assessment index system and weight matrix by measuring the disorder degree in the system. The data requirement of the weight matrix is smaller, and the result is more sensitive to the change of indices [24–27].

Up until now, the index system of the environmental effect assessment of a CEPB lacks determined and completed assessment criteria [28]. To solve the problem, this study established appropriate indices using qualitative and quantitative methods. Then the assessment index system and weight matrix were established based on information entropy and comprehensive AHP. In this method, firstly, the fuzzy membership of the environmental, social and economic effect, as well as environmental management

and sustainability factors, are calculated using the fuzzy membership function. Then the weight value of each level is calculated based on information entropy and comprehensive AHP [29,30]. With the fuzzy mathematical method, effects on environment, society, and economy, as well as environmental management and sustainability factors of a CEPB, are analyzed [31]. Finally, the assessment result of the overall environmental effect of a CEPB can be obtained.

### 2.2.1. Fuzzy Membership Function of Assessment Indices

The most and least optimal values of each index are determined by the development of a CEPB and environmental quality standards of different factors (e.g., water, soil, gas, biology). The fuzzy membership of every index layer is calculated by the fuzzy membership function [31]. Then the membership matrix of environmental effect assessment factors in CEPB areas could be determined by the indices' fuzzy membership.

If the index is positively correlated, the fuzzy membership function is:

$$r_{ij} = \frac{x_{ij} - x_{iMIN}}{x_{iMAX} - x_{iMIN}} \tag{1}$$

If the index is negatively correlated, the fuzzy membership function is:

$$r_{ij} = \frac{x_{iMAX} - x_{ij}}{x_{iMAX} - x_{iMIN}} \tag{2}$$

where $r_{ij}$ represents the relative membership degree of index $i$ to CEPB environmental effects in sample $j$, $x_{ij}$ represents the current index value, $x_{iMIN}$ represents the least optimal value of the same index in different samples and $x_{iMAX}$ represents the most optimal value.

### 2.2.2. Weight Matrix of Assessment Index System of the Environmental Effect in CEPB Based on Information Entropy

There are many methods to determine the weight of the index system, such as principal component analysis [32], deviation maximization method [33], gray correlation analysis [34] and fuzzy synthesis method [35]. However, principal component analysis, deviation maximization method, mean square deviation method [36], artificial neural network and other methods need to calculate index system weight on the basis of mass data statistics, and the grey correlation method and other methods rely on the subjective will of the index system builder too much [37,38]. In this study, entropy weight method was an appropriate method to establish the index system. Entropy weight method has been widely used in determining the weight of this type of multi-objective and multi-level comprehensive evaluation. The weight calculation method based on information entropy is widely used in water resources carrying capacity assessment, geological assessment, aviation comprehensive assessment and medical research.

Information entropy is a measurement of the disorder degree in a system. The greater the index in system changes, the smaller its information entropy is and the more information the index provides. Therefore, the weight value could be calculated by the variation of each index in system.

(1) Standardized index value

$$f_{ij} = \frac{r_{ij}}{\sum\limits_{j}^{n} r_{ij}} \ n \text{ represents the total number of samples} \tag{3}$$

where $f_{ij}$ is the proportion of Sample $j$ by index $i$.

(2) Information entropy of the index $i$

$$H_i = -k \sum\limits_{j=1}^{n} f_{ij} ln f_{ij} \tag{4}$$

where $H_i$ is the entropy of index $i$ if $f_{ij} = 0$, then we assume that $f_{ij} ln f_{ij} = 0$ and $k = 1/ln\, n$, so we can get $0 < H_i < 1$.

(3) Weight of the index $i$ based on information entropy

$$W_i = \frac{1 - H_i}{m - \sum\limits_{i=1}^{m} H_i}, \text{and} \sum\limits_{i=1}^{m} W_i = 1, 0 < W_i < 1, \tag{5}$$

where $W_i$ is the weight of index $i$ based on information entropy, $m$ represents the total number of indices in the assessment index system.

### 2.2.3. The Comprehensive Assessment Model of Environmental Effect of a CEPB

This study establishes the assessment model of the environmental effect of a CEPB based on fuzzy mathematics theory [39]:

$$E = W \times R \tag{6}$$

where $E$ represents the actual assessment result of the environmental effect of a CEPB. $W = (W_i)$ represents the weight matrix of environmental effect index, and $R = (r_{ij})$ represents the fuzzy membership matrix in samples.

For any sample $X' = (x'_i)$, its fuzzy membership matrix is $R' = (r'_1, \ldots, r'_m)$, then the assessment result is:

$$E' = W \times R' = (W_1, \ldots W_m) \begin{bmatrix} r'_1 \\ M \\ r'_m \end{bmatrix} \tag{7}$$

### 2.3. Assessment Framework and Index System and Criterions of Environmental Effect of a CEPB

The assessment index system of the environmental effect of a large CEPB consists of indices which are interrelated, complementary, hierarchical and structural. These indices indicate not only the connection between subsystems but also the characteristics of the whole area. According to the characteristics of the exploitation, construction, management and ecological restoration of CEPBs, as well as the application and practical operation of the assessment indices, there are two ways to assess the indices. One refers to the unified regulations of the relative industries (such as national power, highway or oil industries). The other refers to a selection of the feasible indices according to the characteristics of the CEPB construction project, and establishment of the regional environmental effect assessment index system, as well as the criterion according to the specific situation in the CEPB area and the regulations of other regions [40].

The establishment of the environmental effect assessment index system of a large CEPB is based on the assessment framework (see Figure 2) and environmental impact identification, which is mainly according to the environmental quality standards of all kinds of environmental factors, such as the surface water environmental quality standard (GB2828-2002), ambient air quality standard (GB3095—2012), etc. Additionally, some relevant contents in some standards such as Technical Guidelines and Standards for Environmental Impact Assessment, General Program of Technical Guidelines for Environmental Assessment of Master Planning (HJ450-2008), Pollutant Discharge Standard for Coal Industry (GB20426—2006) and Cleaner Production Standard-Coal Mining Industry (HJ450-2008), etc. [41–45] are also included. The assessment index system is established by theoretical analysis and expert consultation and is based on the actual situation of a CEPB's construction.

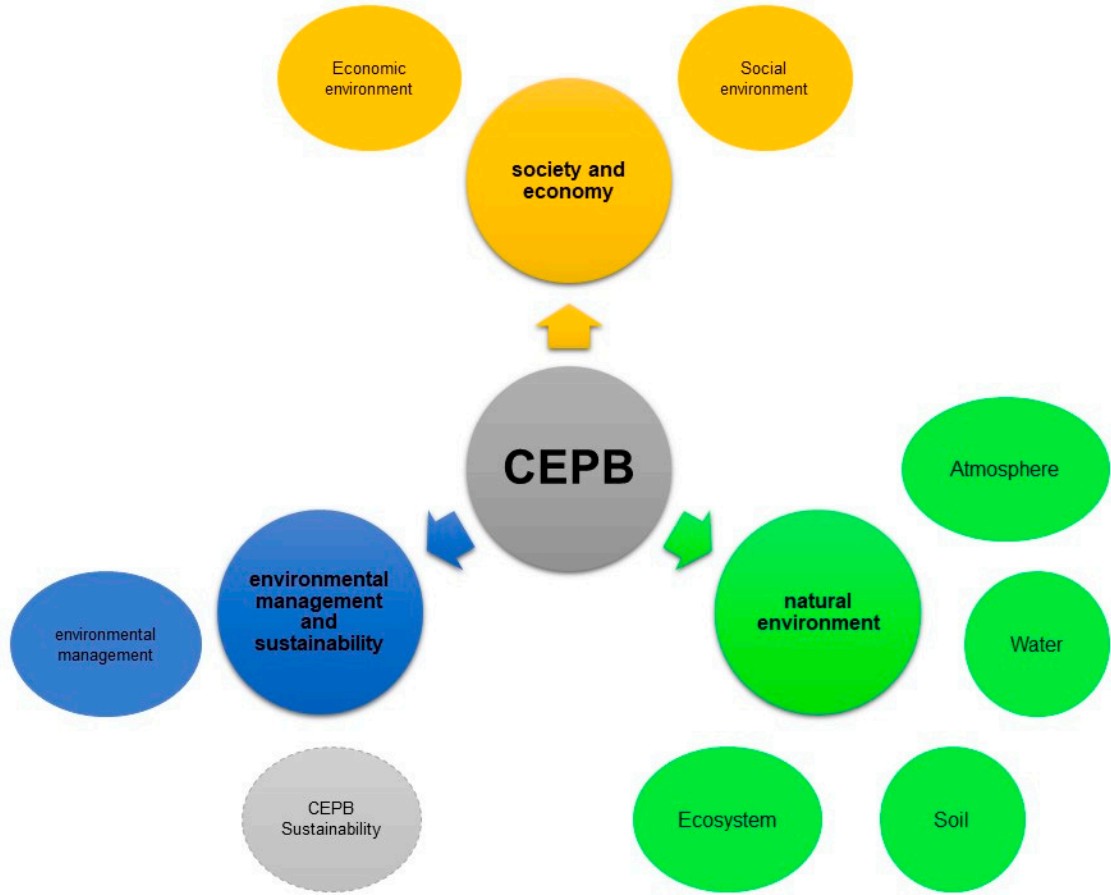

**Figure 2.** Assessment framework.

Due to the lack of a unified assessment criterion of environmental effect of CEPBs, this study divided the environmental effects of CEPBs into six levels based on the practical research and available studies, which were poorer, poor, medium, good, better, excellent [12,16,46]. The index criterion of each level was in accord with relative environmental standards and design criterions as well as the generally accepted standard values of environmental effect indices of CEPBs at home and abroad. Then the assessment indices based on the fuzzy membership function and information entropy model were transferred to the relative standard values of the assessment results, which could be graded and compared to the assessment results. The specific grading criterion is shown in Table 1.

**Table 1.** Grading standard for the integrated effect assessment of a coal-electricity production base (CEPB) in Xilin Gol.

| Constraint Layer | Poorer | Poor | Medium | Good | Better | Excellent | Criterion Layer | Poorer | Poor | Medium | Good | Better | Excellent |
|---|---|---|---|---|---|---|---|---|---|---|---|---|---|
| natural environmental effect | <0.105 | 0.105 ~0.259 | 0.259 ~0.363 | 0.363 ~0.474 | 0.474 ~0.584 | >0.584 | atmospheric environment | <0.084 | 0.084 ~0.112 | 0.112 ~0.12 | 0.120 ~0.121 | 0.121 ~0.122 | >0.122 |
| | | | | | | | water environment | <0.003 | 0.003 ~0.051 | 0.051 ~0.104 | 0.104 ~0.151 | 0.151 ~0.193 | >0.193 |
| | | | | | | | soil environment | <0.018 | 0.018 ~0.059 | 0.026 ~0.072 | 0.072 ~0.111 | 0.111 ~0.163 | >0.163 |
| | | | | | | | ecosystem | - | <0.037 | 0.037 ~0.066 | 0.066 ~0.09 | 0.090 ~0.106 | >0.106 |
| social and economic effect | <0.042 | 0.042 ~0.072 | 0.072 ~0.109 | 0.109 ~0.147 | 0.147 ~0.177 | >0.177 | economic environment | <0.042 | 0.042 ~0.047 | 0.047 ~0.06 | 0.060 ~0.072 | 0.072 ~0.078 | >0.078 |
| | | | | | | | social environment | - | <0.025 | 0.025 ~0.049 | 0.049 ~0.074 | 0.074 ~0.099 | >0.099 |
| environmental management effect and sustainability | - | <0.016 | 0.016 ~0.033 | 0.033 ~0.049 | 0.049 ~0.066 | >0.066 | environmental management | - | <0.016 | 0.016 ~0.033 | 0.033 ~0.049 | 0.049 ~0.066 | >0.066 |

*2.4. Data*

The data needed for the index system of environmental effect assessment of a CEPB are from different sources. The data used in this study includes the Xilin Gol League's TM satellite remote sensing imagery and soil distribution information, as well as related geographic information data [47,48]. The basic data of the natural environmental effect assessment are mainly from the interpretation data of LANDSAT remote sensing images as well as ecological environment indices, meteorological data and environmental monitoring data, etc. Most of the ecosystem indices (e.g., land use types, vegetation cover, soil erosion ratios) are based on the interpretation of LANDSAT remote sensing images and ecological environment indices. Part of the atmospheric environment indices (e.g., air pollution index, climate index and acid rain index) are from the National Meteorological Bureau and the Regional Meteorological Bureau. Water environment indices (e.g., groundwater reserves, water quality, and surface water) are from the regional hydraulic supervision department, and soil environment indices are from a direct monitor. The basic data of social and economic effect including environmental management and sustainability assessment are from the national, regional or local statistical yearbooks and relevant statistical data.

## 3. Results

Based on the information entropy and comprehensive assessment method, the assessment results of the weight value, constraint layer, criterion layer, index layer and key quantitative assessment variables of the comprehensive assessment index system of environmental effect in Xilin Gol CEPB are shown in Table 2. The comprehensive environmental effect of the large CEPB in Xilin Gol was 0.64, which was graded as better according to the assessment criterion. It shows that the regional ecological environment has not been damaged by the exploitation construction of a CEPB.

It can be seen from the index system that the uncertainty of these results mainly comes from the possible errors of the monitoring data and uncertain statistical data [48]. There are also some uncertainties in the weight calculation and result calculation of the entropy method. Furthermore, the fusion process of monitoring data and statistical data are complex, so there are relatively large computational uncertainties.



**Table 2.** Assessment index system, weight and assessment results of CEPB in Xilin Gol.

| Constraint Layer | Weight | Result | Criterion Layer | Weight | Result | Index Layer | Weight | Result | Key Quantitative Assessment Variables | Weight | Result |
|---|---|---|---|---|---|---|---|---|---|---|---|
| natural environmental effect | 0.71 | 0.5036 | atmospheric environment | 0.23 | 0.146 | air pollution index | 0.11 | 0.102 | $SO_2$ | 0.03 | 0.032 |
| | | | | | | | | | $N_xO_x$ | 0.04 | 0.039 |
| | | | | | | | | | pm2.5(24h mean) | 0.02 | 0.012 |
| | | | | | | | | | pm10(24h mean) | 0.02 | 0.019 |
| | | | | | | climate index | 0.06 | 0.044 | rainfall | 0.03 | 0.033 |
| | | | | | | | | | annual mean wind speed | 0.03 | 0.012 |
| | | | | | | acid rain index | 0.06 | 0.000 | rainfall PH(annual mean) | 0.06 | 0.000 |
| | | | water environment | 0.19 | 0.099 | groundwater reserves | 0.03 | 0.020 | groundwater drainage quantity | 0.03 | 0.020 |
| | | | | | | surface water quality | 0.06 | 0.033 | PH | 0.03 | 0.021 |
| | | | | | | | | | COD | 0.03 | 0.012 |
| | | | | | | groundwater quality | 0.10 | 0.046 | permanganate indices | 0.03 | 0.026 |
| | | | | | | | | | total hardness($CaCO_3$) | 0.05 | 0.000 |
| | | | soil environment | 0.18 | 0.181 | soil pollution index | 0.12 | 0.123 | fluoride | 0.02 | 0.020 |
| | | | | | | | | | Pb | 0.03 | 0.031 |
| | | | | | | | | | Cr | 0.03 | 0.030 |
| | | | | | | | | | Hg | 0.02 | 0.025 |
| | | | | | | | | | As | 0.04 | 0.037 |
| | | | | | | soil nutrient index | 0.06 | 0.058 | total nitrogen | 0.03 | 0.028 |
| | | | | | | | | | organic matter | 0.03 | 0.029 |
| | | | ecosystem | 0.11 | 0.078 | land use type index | 0.03 | 0.022 | ratio of land area affected by energy bases | 0.03 | 0.022 |
| | | | | | | vegetation cover index | 0.03 | 0.018 | mine vegetation fractional cover | 0.03 | 0.018 |
| | | | | | | soil loss/erosion index | 0.05 | 0.038 | ratio of increased soil loss/erosion in mining area | 0.03 | 0.020 |
| | | | | | | | | | soil erosion modulus | 0.02 | 0.019 |
| social and economic effect | 0.22 | 0.1012 | economic environment | 0.12 | 0.018 | level of economic development index | 0.04 | 0.009 | GDP per capita | 0.04 | 0.009 |
| | | | | | | economic benefit index | 0.04 | 0.004 | energy consumption per unit of output value | 0.04 | 0.004 |
| | | | | | | circular economy index | 0.04 | 0.005 | proportion of circular economy | 0.04 | 0.005 |
| | | | Social environment | 0.10 | 0.083 | residents' living standard | 0.07 | 0.071 | disposable income per capita | 0.03 | 0.016 |
| | | | | | | | | | Gini index | 0.04 | 0.055 |
| | | | | | | social education index | 0.03 | 0.012 | proportion of bachelors or above in professionals | 0.03 | 0.012 |
| environmental management and sustainability | 0.07 | 0.0352 | environmental management | 0.07 | 0.035 | land reclamation index | 0.03 | 0.026 | ratio of land reclamation in mining areas | 0.03 | 0.026 |
| | | | | | | water resources protection index | 0.04 | 0.009 | Reuse rate of mine water | 0.04 | 0.009 |

## 4. Discussion

### 4.1. Graded Assessment Result of Environmental Effect of a CEPB

Referring to the assessment criterion of the environmental effect of a CEPB, the environmental effect was graded as better, which tells us that the ecological environment is still in a relatively stable and healthy state during the exploitation and construction of a CEPB (see Table 3). The natural environment effect of the study area was graded as better, and effects on society, the economy, and environmental management were graded as good. For the natural environment, the atmosphere and soil environment were graded as excellent, showing the atmospheric and soil environment in the area were hardly influenced by the development of a CEPB and are still in a relatively healthy state. The ecosystem is good, showing that it has been influenced by the CEPB but is still relatively healthy. The environmental effect assessment of water environment was medium, which indicates that the influence on water environment by the CEPB is relatively large. For the social and economic effects, the economic environment was graded as poorer, showing that the economic capability of the study area has not been substantially promoted by the development of a CEPB.

**Table 3.** Graded assessment results of the environmental effect of a CEPB.

| Constraint Layer | Grade | Criterion Layer | Grade |
|---|---|---|---|
| natural environmental effect | better | atmospheric environment | excellent |
| | | water environment | medium |
| | | soil environment | excellent |
| | | ecosystem | good |
| social and economic effect | good | economic environment | poorer |
| | | social environment | better |
| environmental management and sustainability | good | environmental management | good |

### 4.2. Natural Environment Effect

For the assessment result of the natural environment effect, the soil environment was the highest while ecosystem was the lowest, as shown in Figure 3. Referring to the graded criteria, the environmental effect of atmospheric and soil environment were both graded as excellent and the ecosystem was good, while the water environment was medium. According to the assessment result of the index layer, only the scores for the air pollution index and soil pollution index exceeded 0.1, showing that the environmental effect indices about atmosphere and soil perform better.

### 4.3. Social and Economic Effect

The social and economic effect of the CEPB in the study area was graded as good. The assessment result about the index layer of social and economic effect and key quantitative assessment variables of the CEPB are shown in Figure 4. The social environment effect performed better than the economic environment. As for social and economic effect assessment indices, residents' living standard was significantly higher than the others. Among all the key quantitative assessment variables, Gini index was the highest, with obvious advantages over the others.

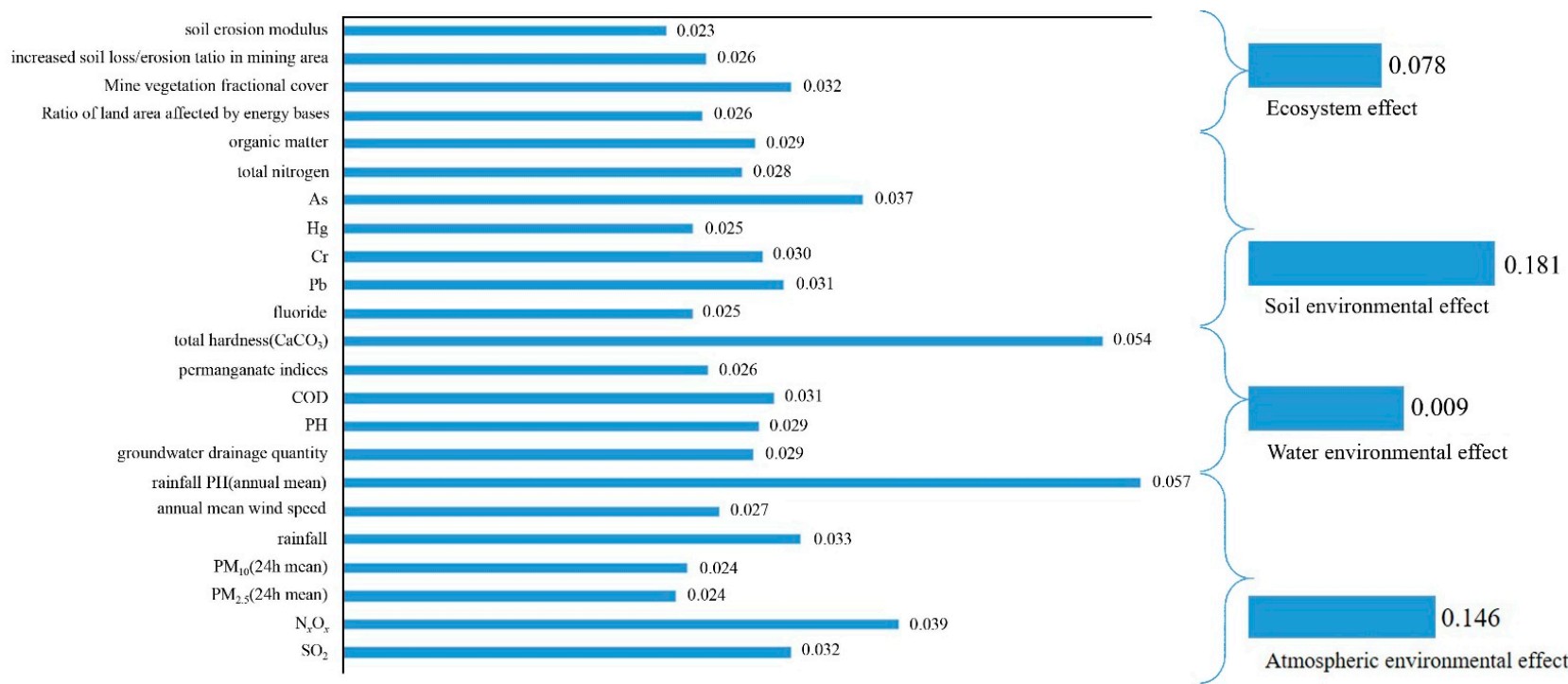

**Figure 3.** Assessment results of environmental effect of a CEPB in the study area.

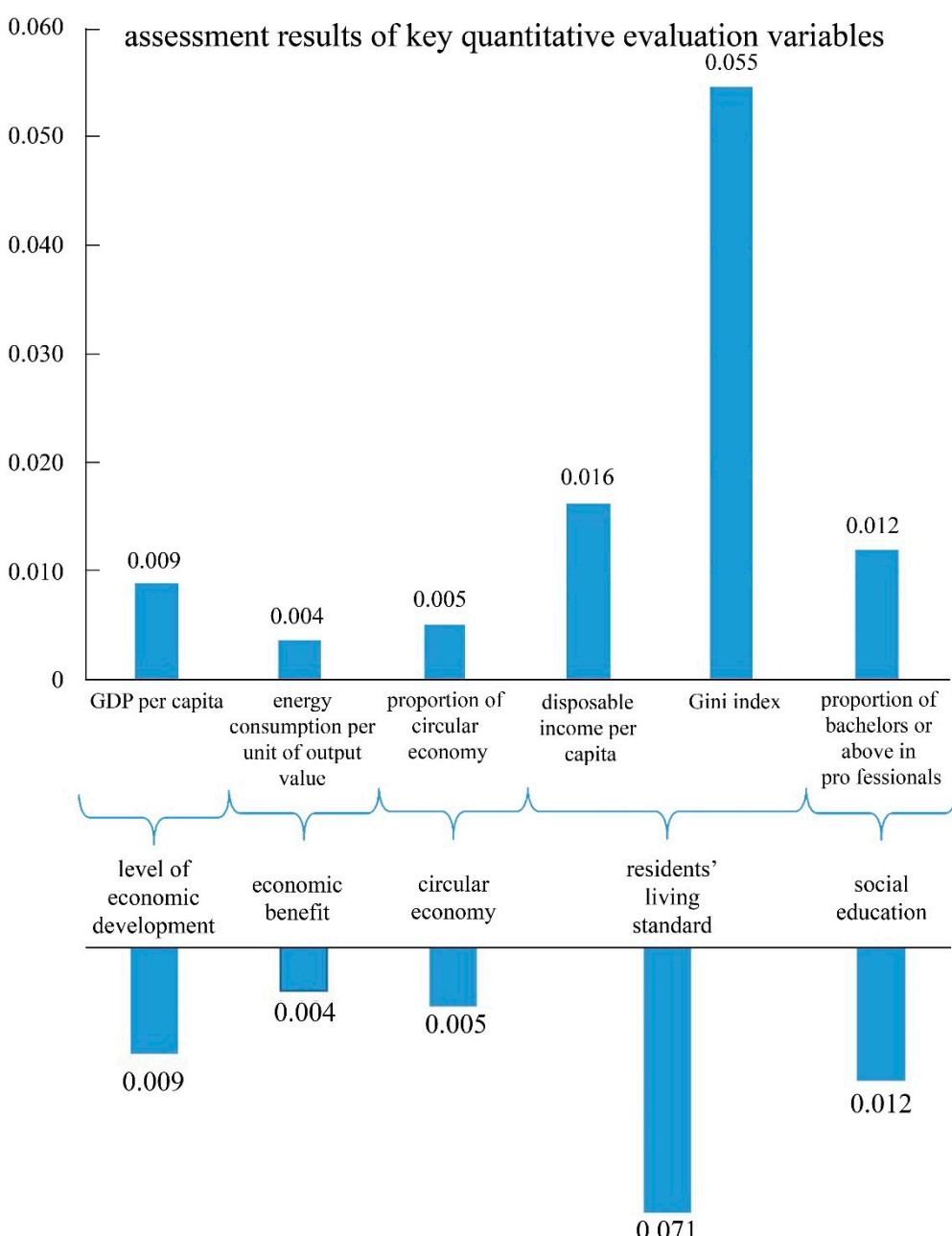

**Figure 4.** Assessment result of the social and economic effect of a CEPB in the study area.

*4.4. Environmental Management and Sustainability*

The assessment of environmental management and sustainability play an important role in the comprehensive environmental effect of a CEPB. The assessment result of the study area was graded as good. The assessment result of the land reclamation index was 0.26, while water resources protection was 0.09, which indicates that more progress has been made in land reclamation than water resources protection in the study area.

## 5. Development Strategy towards Sustainability

*5.1. Establish Comprehensive Framework for Water Sustainable Development*

Water environment and water resources have been important factors in maintaining ecological balance and environmental capacity, also improving the ecological environment in the construction of

CEPB s. In the process of planning, construction and operation of CFPBs, it is necessary to establish a comprehensive framework to ensure sustainable development of water resources as this can solve the contradiction between the supply and demand of water resources, regulate pollution discharge and reduce water ecological damage. In the planning stage of a CFPBs' construction, the assessment of impact on water resources and estimation of water environment capacity should be an important evaluation index of planning rationality. Through the comprehensive management and regulation of water price, the government urges the coal power base to take the initiative to save water and improve the efficiency of water production. The CFPBs continuously increase the construction of water-saving facilities, improve the capability of water-saving technology and enhance the efficiency of water production so as to save water resources and improve water quality. In addition, the establishment of a strict water pollution monitoring system requires CFPBs to continuously improve the capacity of sewage treatment, which not only reduces water pollution and water environmental risks, but also contributes to the reuse of water resources.

*5.2. Establish Process Monitoring Mechanisms*

The government and CFPBs must specify the indicators and elements that need long-term monitoring [21], including operation parameters of a coal power base, environmental pollution sources, ecological risk sources and so on. Monitoring environmental pollution sources requires the establishment of pollutant discharge monitoring and the determination of threshold values of factors and indicators based on a social environmental monitoring network. The monitoring of ecological risk sources should focus on the possible ecological risks and types of ecological damage in the coal power base. The monitoring of ecological risk sources should be oriented by regional ecological security and establish a long-term quantitative monitoring system.

*5.3. Devise Metrics of Sustainability and Keep Regular Assessment*

It is necessary to establish a sustainability evaluation index system and corresponding quantitative evaluation method system based on the characteristics of CFPBs' development [12]. The construction of the index system should be based on the index system of environmental impact assessment. The index system should not only focus on the ecological environmental problems caused by the construction and operation of CFPBs, such as ecological damage and environmental pollution, but also needs to comprehensively consider the process and mechanism of impacts of CFPBs on the surrounding cities, social residents and the economic system. The CFPBs' sustainability needs to be quantitatively assessed on a regular basis by inviting third-party assessments. The assessment process of a CFPBs' sustainability summarizes the sustainability status and changing process over a period of time and identifies the key elements and key processes.

*5.4. Fine Management to Continuously Decrease the Pollutant in Production Processes*

Fine management can be applied to reduce pollution emission and environmental impacts of the production process. In the process of construction and operation of CFPBs, the implementation of fine management can make the chain of production process clear. For instance, a kanban can be prepared to note down the environmental impact of every link in the whole production process. Kanban is a technical term in management which can be understood as a dashboard that records every important note in a production process for reminding. By doing so, the total amount of pollution emissions and the process of environmental impact can be reasonably separated into each production link to ensure the refined control of pollution emissions and environmental impact in each production process.

*5.5. Meet Regional Sustainable Development Goals*

The construction and operation of a CEPB should meet goals of regional sustainable development and ecological environmental plans. The decision-maker should pay more attention to the regional

ecosystem, focus on the objectives of ecosystems and protection of water resources, and determine CFPB actions based on ecological processes and responsive mechanisms to achieve sustainable development.

## 6. Conclusions

Coal currently accounts for nearly 70% of the energy consumption in China, and it is expected to be a vital energy source in the near future. Planning and construction of national large-scale CEPBs are an important aspect of China's 12th coal industry five-year plan and the development strategy for future energy. CEPBs have huge potential of electricity generation and are important for maintaining energy security and structure. However, the construction and development of CEPBs have resulted in local ecological and environmental problems. The sustainable development of CEPBs is becoming increasingly important. The result of this study shows that the method based on information entropy and comprehensive AHP could be applied to the assessment of environmental effects of large CEPBs, and the result has high sensitivity and significant variations. The method in this study can be applied to different assessment objects in different areas of different spatial scales. It can be used on the assessment of CEPBs in multiple regions of China, and the time series assessment of a certain CEPB as well. This method can help to understand the key issues in the exploitation, construction and operation of CEPBs.

The environmental effect of a CEPB in Xilin Gol according to the method based on information entropy and comprehensive AHP is 0.64, showing that the study area is in good condition. For constraint layers, the natural environment effect is graded as better, and the social and economic effect and environmental management and sustainability are good. For the natural environment effect, the result of the soil environment is the highest while the ecosystem is the lowest. According to the grading criteria for environmental effect assessment, atmospheric and soil environment are both graded as excellent, while the water environment is medium. The water environment is an important issue in the development of a CEPB in Xilin Gol. The assessment results of social and economic effect is graded as good, and residents' living standard is significantly higher than other indices. For environmental management and sustainability, more progress has been made in land reclamation than protection of water resources in Xilin Gol.

According to the comprehensive assessment and grading of the environmental effect of a CEPB in Xilin Gol, water environment is the key issue to be considered during the development of a CEPB. Both the effort and achievement of environmental management and sustainability measures in water protection are relatively weak, which should be promoted in future development. Based on the assessment of the environmental effect of the Xinlin Gol case, this paper puts forward five development strategies for CFPB sustainability.

**Author Contributions:** C.W. and Y.Y. conceived and designed the experiments; S.L. and G.W. performed the experiments; J.L. analyzed the data; G.W. contributed materials tools; C.W., Y.Y. and Y.Q. wrote the paper. All authors have read and agreed to the published version of the manuscript.

**Funding:** This study was supported by the National Key Research and Development Program (No. 2019YFB2102004) of China.

**Acknowledgments:** This study was strongly supported by local leaders of Xilin gol League and the head of the coal power base company. And thanks to Shenhua Shengli Coal Mine, Baiyinhua Coal Power Group and other local Coal-Power companies for the supporting of data acquisition.

**Conflicts of Interest:** The authors declare no conflict of interest.

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
