# Peer review of "Integrated Environmental Effect Assessment on Large Coal-Electricity Production Base and Sustainability Strategy: A Case in Xilin Gol, China"

_sustainability, doi:10.3390/su12155943_

Round 1
Reviewer 1 Report
The paper is "appeal" to a practicing industrial engineer.
The title is attractive and interesting.
The abstract is clear and to the point, stressing both the specific application and the generic aspects of the work.
The paper is correct technically and consistently written.
The paper is intelligible, and relevant, to practicing professional engineers.
The figures are clear explicit and properly labeled.
The references are complete, and relatively easy to obtain.
The submitted paper is well prepared for publication in Sustainability.
I suggest to accept.
Major revision:
Please, add original and newest papers to the list of references that deal with the meaning of information entropy. For example:
[1] C. Shannon. A mathematical theory of communication. Bell System Technical Journal 27(3):379-423 and 623-656, 1948. DOI:10.1109/9780470544242.ch1.
[2] C. Shannon, W. Weaver. The Mathematical Theory of Communication. The University of Illinois Press, Urbana IL, 1964. DOI:10.2307/3611062.
[3] M. Belies, S. Guiasu. A quantitative-qualitative measure of information in cybernetic systems. In IEEE Trans. Inf. Theory IT-4, pp. 593-594. 1968. DOI:10.1109/tit.1968.1054185.
[4] E. T. Jaynes. Information theory and statistical mechanics. Phys Rev 106(4):620-630, 1957. DOI:10.1103/physrev.108.171.
[5] A. Delgado-Bonal, J.Martín-Torres. Human vision is determined based on information theory. Scientific Reports 6(1), 2016. DOI:10.1038/srep36038.
[6] J. Shore, R. Iohnson. Axiomatic derivation of the principle of maximum entropy and the principle of minimum cross-entropy. IEEE Trans Inf Theory 26(1):26-37, 1980. DOI:10.1109/tit.1980.1056144.
[7] M. Donald. On the relative entropy. Commun Math Phys 105:13-34, 1986. DOI:10.1007/bf01212339.
[8] L. R. Nemzer. Shannon information entropy in the canonical genetic code. Journal of Theoretical Biology 415:158-170, 2017. DOI:10.1016/j.jtbi.2016.12.010.
[9] S. Yu, T.-Z. Huang, X. Liu, W. Chen. Information measures based on fractional calculus. Inf Process Lett 112(23):916-921, 2012. DOI:10.1016/j.ipl.2012.08.019.
[10] S. Yu, T.-Z. Huang. Exponential weighted entropy and exponential weighted mutual information. Neurocomputing 249:86-94, 2017. DOI:10.1016/j.neucom.2017.03.075.
[11] K. Krechmer. Relational measurements and uncertainty. Measurement 93:36-40, 2016. DOI:10.1016/j.measurement.2016.06.058.
Author Response
The authors would like to thank the reviewer #1 for his/her comments and suggestions. We will carefully revise the suggestions of reviewer #1. As suggested in reviewer #1’s comment, we have added these references in above list in the right place in our manuscript. And the reference format met the demand of this journal.

Reviewer 2 Report
This paper is not very well written and needs extensive editing. You must also better describe the derivation of the indices (not indexes) especially explaining fuzzy mathematical modeling. In your results section you should be more explicit in tying together the specific models with results. It is rather scattered and is difficult to read.
You also did not discuss sustainability policies for CEPB for Chinese policy makers.
Author Response
We are appreciated the reviewer #1. For improving this manuscript, we have revised several section and the English expression. Also, we have invited native English professional person to make comprehensive revision of the grammar and spelling of this manuscript.
(1) We have revised the description and expression in section 2.3 to discuss in depth the derivation of the indices and the significance to the sustainable development of coal power base, and added the corresponding literature for support. The incorrect word
‘indexes’ was corrected also. In section 2.2, we have added more description and references to explain the fuzzy mathematical modeling.
(2) As the reviewer #2’s comment, we have rewritten the description in section 3 Result, to make more specific analysis on the model and the result.
(3) In terms of the logicality and readability of this manuscript, we have also made a lot of modifications. We have reorganized the logic of the article in many section, and adjusted and improved the expression and English language, so as to further improve the logicality of the article and make it easier to understand.
You also did not discuss sustainability policies for CEPB for Chinese policy makers.
(4) We are appreciated the comment about the sustainability policies for Chinses policy makers. As this comment, we have added one more development strategy in the section 5 and deeply discussed how should the government decision makers consider the overall sustainability of the CEPBs and the regions in which they are located.

Reviewer 3 Report
The main purpose of this research is to assess the environmental effect of large scale Coal-Electricity Production Base (CEPB) in China, based on information entropy and analytic hierarchy process. Moreover, the authors quantitatively evaluate the environmental effects of large-scale CEPB in Xilin Gol, Inner Mongolia. From the overall presentation I would say that an interesting research work has been done.
However, many concerns arise regarding the manuscript.
General comments
- The research questions as well as the original contribution of the work, comparing to other previous works are not adequately presented.
- From a methodological point of view, a more clear explanation of the analysis needs to be provided.
- A more extensive and critical literature review of the research topic should be added.
- A further theoretical analysis of the applied model is needed.
- A more in-depth discussion of the results needs to be provided in relation to the theoretical analysis.
- The discussion of the uncertainty of the results is missing from the analysis.
- The language of the manuscript needs to be improved, since many sentences are not well expressed. Grammatical and syntax errors should be corrected.
- All the acronyms should be explained within the manuscript.
- All the equation symbols must be defined within the manuscript.
- All the equations should be numbered.
Additional specific comments and recommendations for the improvement of the manuscript:
Abstract
[Line 17] “… development, Coal-electricity production base (CEPB) are integrated plants comprise coal mining…” instead of “…development. CEPB are integrated plants comprise coal mining…”.
- Introduction
General note: In this section, the authors could refer to their original contribution to the analysis of the problem.
[Line 37] “available power coals”?
- Materials and Methods
General note: Section 2.1 is too short. More information is needed about the study area.
[Fig. 1] Legend: font size must be increased.
[Lines 111-112] The fuzzy membership function need to be further explained.
[Lines 122-128] The parameters of the model need to be further explained.
[Lines 165-166] “…the Environmental effects of CEPB into six levels based on the practical research and available studies…”. A reference is required here.
[Table 1] The results presented in Table 1 should be further explained within the manuscript.
[Lines 174-186] Section 2.4: References are needed in this section.
- Results
[Table 2] The results presented in Table 2 should be further explained within the manuscript. The sum of the weights should be checked. In addition, Gini index needs to be explained.
- Discussion
General note: This section should be further analyzed according to the calculation results.
[Fig. 3] Figure 3 is not clear. Font size should be increased.
[Fig. 4] The quality of the figure should be improved.
- Development strategy towards sustainability
[Lines 239-242] “In the process of planning, construction and operation …water ecological damage and damage”. Please correct the sentence.
[Lines 273-274] “Fine management to reduce pollution emission and environmental impact in the production process”. Please correct the sentence.
Conclusion
General note: In this section, the original contribution of the research has to be presented by focusing on the research results based on the research questions.
Author Response
We are appreciated the comments from reviewer #3. He/She put forward very rich and detailed suggestions for the revision of this manuscript to help us improve the overall quality and details of this manuscript. We have fully revised and improved the manuscript according to the requirements of reviewer #3. Once again, we thank the reviewer #3 for his/her thoughtful comments. The specific responses could see the attachment cover letter.

Reviewer 4 Report
The authors deal with an interesting topic of construction project management by researching the possibilities to increase the sustainability of projects related to large coal-electricity production base (CEPB). The approach is interesting as the basic idea is to make a comprehensive assessment of CEPBs with the environment effect assessment approach. Although the topic is interesting, several major insufficiencies need to be improved.
Suggestions for improvement:
- The manuscript should be set according to the Journal’s template and instruction to authors (text, figures, tables, equations, references, etc.)
- Check and improve the English language and grammar throughout the paper (check misspellings, writing in the first person, etc.), as well as all figures and tables (both must be readable)
- Abstract should be rewritten to provide necessary elements such as research questions, goals, hypotheses, methods, results, and conclusions. Now it is too long and unclear
- The introduction does not provide sufficient background and includes all relevant references. The used references are not novel and some fundamental references are missing as well as the recent ones considering the research problem. The authors should be consistent in writing. The research problem is clear but research goals and hypotheses are not clearly stated
- The literature review should be improved. At the moment this section lacks a critical overview of the other approaches in solving stated research problem and the methodology upgrade that is proposed by this research
- The research design is sound but not clearly written. The research methodology should be clear and the hows and the whys of used methods should be clearly visible
- Additional clarification should be added for given assessment framework because Fig 2. offers 8 criteria, while results are given for only 7 of them
- Explanations of results and their discussion are clear, but the discussion about the research significance is missing. It is not clear how the proposed procedure is validated. Also, add some additional discussion of findings in relation to the research framework as well as research goals and hypotheses are needed
- The authors are urged to draw conclusions that are more specific. At the moment it seems like good observations and arguments that are currently missing from the discussion section. There should be a clear connection with the research problem, goals, and results
Overall, I strongly urge the Author to reconsider the above-mentioned comments, rewrite the paper accordingly, and resubmit.
Author Response
We thank the appreciation of this manuscript from reviewer #4. We have carefully revised all kinds of comments proposed by reviewer #4, so that the overall level of this manuscript has been improved by leaps and bounds. The specific responses could see the attachment of cover letter.

Round 2
Reviewer 2 Report
Much better however I detected several grammatical errors.
Please edit the English more carefully.
Author Response
The authors would like to thank the reviewer #2 for his/her comments and suggestions. We will carefully revise the suggestions of reviewer #2.
Reviewer 3 Report
In the revised edition, the manuscript has been improved.
- Further editing is needed.
- Please check the sum of the weights in Table 2 (0.72+0.22+0.07,…).
- Figure 3 is not clear.
Author Response
The Reviewer #3’s comments:
- Further editing is needed
Response: We are appreciated the reviewer #3. For improving this manuscript, we have revised several section and the English expression. Also, we have invited native English professional person to make comprehensive revision of the grammar and spelling of this manuscript.
2 Please check the sum of the weights in Table 2 (0.72+0.22+0.07,…).
Response: We are appreciated the comment about the mistake in Table 2. We have check all the weight in Table 2, and found that a weight number in Table 2 was miswritten while writing the article, which caused this problem. We have changed the number based on the original data table. Further, we have checked all the data results in the full text to make sure there were no similar clerical errors.
3 Figure 3 is not clear.
Response: We have updated the Figure 3 and increased the font size in this figure in the former revised version. However, when the image is copied into the article, the system reduced the accuracy automatically so that the image remains unclear. We have adjusted the image accuracy and checked the sharpness of all the other images again.
Reviewer 4 Report
In revised version authors gave additional insights into their research and also acted upon given comments and suggestions, and gave all required clarifications. Overall, I believe that the article provides valuable content to the present body-of-knowledge.
Author Response
The authors would like to thank the reviewer #4 for his/her comments and suggestions. We will carefully revise the suggestions of reviewer #4.